Co-applied biochar and drought tolerant PGPRs induced more improvement in soil quality and wheat production than their individual applications under drought conditions

Malik Laraib 1
Hussain Sabir 1
Shahid Muhammad 2
Mahmood Faisal 1
Ali Hayssam M. 3
Malik Mehreen 4
Sanaullah Muhammad 4
Zahid Zubda 5
Shahzad Tanvir tanvirshahzad@gcuf.edu.pk hereistanvir@gmail.com 1
1 Department of Environmental Sciences, Government College University , Faisalabad , Pakistan
2 Department of Bioinformatics and Biotechnology, Government College University, Faisalabad , Faisalabad , Pakistan
3 Department of Botany and Microbiology, College of Science, King Saud University , Riyadh , Saudi Arabia
4 Institute of Soil and Environmental Sciences, University of Agriculture , Faisalabad , Pakistan
5 Department of Agro environmental Chemistry and Plant Nutrition, Faculty of Agrobiology, Food, and Natural Resources, Czech University of Life Sciences , Prague , Czech Republic
Dąbrowski Piotr
Electronic publication date: 2024 Oct 25
Publication date: 2024
Volume: 12
Electronic Location ID: e18171
Received 2024 Jan 23; Accepted 2024 Sep 3
Copyright: ©2024 Malik et al.
Copyright year: 2024
Copyright holder: Malik et al.
License: This is an open access article distributed under the terms of the Creative Commons Attribution License, which permits unrestricted use, distribution, reproduction and adaptation in any medium and for any purpose provided that it is properly attributed. For attribution, the original author(s), title, publication source (PeerJ) and either DOI or URL of the article must be cited.
License URL: https://creativecommons.org/licenses/by/4.0/

Keywords: Drought, Biochar, PGPR, Soil enzymatic activity, Grain yield, Plant available phosphorus

Funding: Researchers Supporting Project RSP2024R123 This work was funded by the Researchers Supporting Project number (RSP2024R123), King Saud University, Riyadh, Saudi Arabia. The funders had no role in study design, data collection and analysis, decision to publish, or preparation of the manuscript.

==============================
Background

Plant growth and development can be greatly impacted by drought stress. Suitable plant growth promoting rhizobacteria (PGPR) or biochar (BC) application has been shown to alleviate drought stress for plants. However, their co-application has not been extensively explored in this regard.

Methods

We isolated bacterial strains from rhizospheric soils of plants from arid soils and characterized them for plant growth promoting characteristics like IAA production and phosphate solubilization as well as for drought tolerance. Three bacterial strains or so called PGPRs, identified as Bacillus thuringiensis, Bacillus tropicus, and Bacillus paramycoides based on their 16S rRNA, were screened for further experiments. Wheat was grown on normal, where soil moisture was maintained at 75% of water holding capacity (WHC), and induced-drought (25% WHC) stressed soil in pots. PGPRs were applied alone or in combination with a biochar derived from pyrolysis of tree wood.

Results

Drought stress substantially inhibited wheat growth. However, biochar addition under stressed conditions significantly improved the wheat growth and productivity. Briefly, it increased straw yield by 25%, 100-grain weight by 15% and grain yield by 10% compared to the control. Moreover, co-application of biochar with PGPRs B. thuringiensis, B. tropicus and B. paramycoides further enhanced straw yield by 37–41%, 100-grain weight by 30–36%, and grain yield by 22–22.57%, respectively. The co-application also enhanced soil quality by increasing plant-available phosphorus by 4–31%, microbial biomass by 33–45%, and soil K+/Na+ ratio by 41–44%.

Conclusion

Co-application of PGPRs and biochar alleviated plant drought stress by improving nutrient availability and absorption. Acting as a nutrient reservoir, biochar worked alongside PGPRs, who solubilized nutrients from the former and promoted wheat growth. We recommend that the co-application of suitable PGPRs and biochar is a better technology to produce wheat under drought conditions than using these enhancers separately.

Introduction

Drought stress is one of the major abiotic stresses for agriculture particularly in semi-arid and arid areas around the world (Pandey et al., 2017; Egamberdieva et al., 2019). Global demand for water for agriculture is expected to increase by 60% by 2025. Hence, the impacts of drought on agriculture are likely to be further worsened by the reduction of water sources and the growing need for food for fast increasing global population (Boretti & Rosa, 2019). Approximately 80 to 95% of the plant’s total fresh biomass consists of water, and this water is essential for numerous physiological activities such as various aspects of plant growth, development, and metabolism. Drought stress causes disruption of physiological processes, low amounts of nutrients, poor photosynthesis and limited water supply, severely affecting crop growth and output (Danish & Zafar-ul-hye, 2020; Seleiman et al., 2021).

Plant growth promoting rhizobacteria (PGPR), as the name indicates, are bacteria living in rhizosphere of plants and promote plant growth through a multitude of direct and indirect mechanisms. They may, for instance, secrete growth stimulating hormones like indole acetic acid, turn immobile nutrients into available forms like solubilization of phosphate into available phosphorus, directly fix atmospheric nitrogen into ammoniac forms that is easily taken up by plants etc. (Rashid et al., 2016). Furthermore, they also help plants circumvent different biotic stressors like diseases and abiotic stressors like salinity, drought, toxic elements etc. (De Andrade et al., 2023). For instance, in order to impart drought tolerance to plants, they perform a suite of functions like secretion of exopolysaccharides that prevent desiccation, ACC-deaminase, volatile compounds, accumulation of osmolytes, antioxidants, and up- or down-regulation of stress-related genes and influencing a root architecture that’s more suitable for nutrient and water acquisition (Vurukonda et al., 2016; Anli et al., 2020). Hence, they are an effective tool to alleviate drought stress for plants.

Biochar, characterized by its carbon-rich composition, is a stable solid substance formed via the pyrolysis of organic biomass, usually under oxygen-limited conditions (Lehmann, Gaunt & Rondon, 2006). It has attracted interest recently because of its prospective uses in agricultural and environmental management, notably its function in stressed environments (Liang et al., 2014; Akhtar, Andersen & Liu, 2015; Malik et al., 2022). Its porous structure and extensive surface area enable it to trap water and essential nutrients in the soil, thus enhancing both water retention and nutrient accessibility for plants during periods of water scarcity (Biederman & Harpole, 2013). For instance, biochar addition increases available water by 4 to 130%, reducing the saturated hydraulic conductivity in coarse-textured while increasing the same in fine-textured soils (Blanco-Canqui, 2017). Generally, by decreasing the bulk density of soils by 3 to 31% and increasing the soil porosity by 14 to 64%, biochar application substantially improves water infiltration as well as water retention capacity of soils (Devereux, Sturrock & Mooney, 2013; Githinji, 2014). Generally, the improvement in water retention after application of biochar is more significant for coarse-textured soils, whereas little improvement in this regard is found in fine textured soils though the runoff is considerably reduced in the latter (Edeh, Mašek & Buss, 2020; Razzaghi, Obour & Arthur, 2020). Moreover, biochar encourages the growth of beneficial microbial communities in the rhizosphere, which in turn fosters nutrient absorption and enhances plants’ ability to tolerate stress (Bolan et al., 2023).

Biochar and PGPR, being two different types of enhancers of soil quality as well as plant growth, may act synergistically or in combination to further improve soil quality and plant growth by complementing each other’s role. For instance, they have been shown to reduce the need of chemical fertilizers where the biochar served as nutrient reserve and PGPRs served as miners of nutrients trapped in biochar thereby releasing it into the soil for plant uptake (Ijaz et al., 2019; Azeem et al., 2022). Their co-application to saline soils has also been shown to improve soil quality and plant growth where biochar adsorbs the Na+ ions from soil thereby improving K+/Na+ ratio while the PGPRs facilitate uptake of nutrients accrued from biochar in addition to releasing phytohormones thereby stimulating plant growth (Fazal & Bano, 2016; Malik et al., 2024). On the same pattern, it can be hypothesized that the co-application of PGPR and biochar may alleviate drought stress for plants significantly higher than their sole applications due to their complementing functioning.

We designed this study to assess the potential of co-application of biochar and suitable drought-tolerant PGPR to alleviate drought stress for plants, increase their growth and improve soil quality. We isolated bacterial strains from rhizospheric soils from arid regions and characterized them for drought tolerance and plant growth promotion. Suitable PGPRs and wooden tree derived biochar were then applied to wheat grown on moisture deficit soils in a pot experiment. We assumed that when used independently, biochar and drought-tolerant PGPRs would improve soil health indicators and wheat growth and productivity. The second assumption is that the co-application of PGPRs and biochar would improve soil health as well as wheat growth and productivity in a significantly higher amount than their sole application.

Materials and Methods

Sampling of rhizospheric soils for bacterial isolation

Flowering barley (Hordeum vulgare), chickpea (Cicer arietinum), and castor (Ricinus communis) plants cultivated in the arid and semi-arid regions of Punjab, Pakistan, specifically Thal (33.3693°N, 70.5443°E), Layyah (30.9693°N, 70.9428°E), and Bhakkar (31.6082°N, 71.0854°E) were selected to collect rhizosphere soils for isolation of rhizobacteria. Plants were uprooted and gently shaken to remove the loosely attached soil with the roots, while the soil still adhered to roots was collected. In these areas, the soil is sandy and precipitation and irrigation water are scarce. Therefore, the aforementioned drought-resistant crops are grown whenever there is seasonal rain. We assumed that the bacterial strains isolated from rhizosphere of these plants would naturally be drough- tolerant. These soil samples from the rhizosphere were carefully collected and placed in sterilized polythene bags before being transported to the laboratory. They were stored in a refrigerator at 4 °C and subsequently transferred to the laboratory for the purpose of isolating bacterial strains.

Isolation of drought-tolerant rhizobacteria

The rhizosphere soil was collected under sterile conditions and transformed into a soil suspension to isolate rhizobacteria. To isolate bacteria with drought-tolerant characteristics, the dilution plate technique was applied, utilizing nutrient agar (NA) medium supplemented with PEG 6000 to induce drought stress conditions (Fischer et al., 2007). In this process, the rhizospheric soil suspension was diluted using sterilized distilled water. From each dilution, 1 milliliter (1 mL) of the suspension was added to nutrient agar petri dishes (which were also sterilized) containing 3% PEG 6000 for simulating drought tolerance conditions. Subsequently, these petri dishes were placed in an incubator set at 28 ± 1  °C to facilitate bacterial growth. After a 24-hour incubation period, individual bacterial strains were streaked onto fresh plates using the spread plate technique for further purification (Somasegaran & Hoben, 1994).

Plant growth promoting (PGP) characteristics

The plant growth-promoting characteristics of all bacterial isolates were assessed based on the following criteria i.e., indole acetic acid (IAA) production and phosphate solubilization test i.e., halo zone appearance.

Indole acetic acid (IAA) production

IAA production was assessed following the methodology outlined by Chrastil (1976). Briefly, tryptophan was utilized as a precursor for IAA production in NA media. NA media with and without tryptophan were utilized for cultivating bacterial cultures, which were then subjected to incubation in a shaking incubator at 28 °C. Following 24, 48, and 72 h of bacterial growth, one mL of fully matured bacterial cultures (maximum population density) underwent centrifugation at 8,000 rpm and 4 °C for 5 min. Subsequently, 1 mL of the centrifuged supernatant was mixed with two mL of Salkowski’s reagent (composed of 10.8M H2SO4 and 4.5 g FeCl3 in 1,000 mL of distilled water) within a test tube. This mixture was then incubated for 20–25 min to facilitate the IAA assay, with any color change (indicating the formation of a pink auxin complex) being observed at 530 nm using a UV-visible spectrophotometer. The quantification of IAA involved the application of a standard calibration curve, established through linear regression analysis.

Phosphate solubilization test: Halo zone appearance

The ability of the cultures to solubilize phosphate was evaluated using the approach described by Goldstein (1986). Cultures were cultivated in Pikovaskaya’s agar medium containing tri-calcium phosphate as the inorganic phosphate source and incubated for 6 days at 28 °C. During this incubation, a distinct halo zone, or clearing zone, emerged around the bacterial colonies on the Pikovaskaya’s agar. The presence of such a halo zone indicated the successful solubilization of phosphate and was regarded as a positive outcome for phosphate solubilization. These halo zones were quantified using a measuring scale, and subsequently, the phosphate solubilization index (PSI) was calculated for each phosphate-solubilizing strain, employing the formula reported by Fankem et al. (2006).

PSI = total diameter (colony + clear zone)/diameter of colony.

16S rRNA amplification and sequencing

For the identification of the most potent bacterial strains, 16S rRNA gene was amplified and sequenced. The purification and sequencing of the resulting 16S rRNA products were carried out in collaboration with Macrogen in South Korea. The obtained 16S rRNA sequences of the bacterial strains were subsequently compared with existing nucleotide sequences of other strains utilizing BLAST (https://blast.ncbi.nlm.nih.gov/Blast.cgi). Following this, phylogenetic trees were constructed through multiple alignments facilitated by ClustalX and Mega7. Furthermore, data processing was done by using NJ plot for neighbor joining method.

Inoculum preparation

Strains that demonstrated the capacity to produce substantial amounts of IAA and solubilize rock phosphate (insoluble phosphate) in moisture deficit conditions were chosen for the purpose of inoculation. The chosen strains were streaked onto NA media supplemented with 3% PEG 6000 and subjected to incubation at 28 °C for a duration of 2 days. Following this incubation period, bacterial cultures were cultivated utilizing nutrient broth. These cultures were subsequently utilized to inoculate wheat seeds before sowing.

Biochar preparation and analysis

Biochar was produced by pyrolyzing shisham (Dalbergia sissoo) wood trees. The wood, which had been air-dried, underwent pyrolysis within a stainless steel furnace following the methodology outlined by Lehmann et al. (2011). This furnace had a capacity of 10 kg and was equipped with a gas burner designed for this purpose, utilizing natural gas for ignition. The wood was steadily heated at 450 °C for a mean residence time of two hours. Upon completion of the pyrolysis, the resulting material, known as biochar, was allowed to cool. Subsequently, the biochar underwent crushing and was sifted through a 2 mm sieve before being stored in airtight containers for later use. The pH and electrical conductivity (EC) of the biochar were determined using pH and EC meters, employing a weight-to-distilled water ratio of 1:20. Total nitrogen was determined using an elemental analyzer, while total and available phosphorus levels were determined using a spectrophotometer. Additionally, the content of available potassium (K) and sodium (Na) in the biochar was measured using a flame photometer.

Pot experiment

The potential of co-application of the biochar and selected strains to boost wheat growth and yield under moisture deficit conditions was assessed in a pot trial conducted in the glasshouse of the University of Agriculture Faisalabad, which is located in a subtropical area of Pakistan (31.4294°N, 73.0750°E) at an elevation of 605 ft.

The physico-chemical attributes of the soil used in the experiment were determined after sieving it through a two mm mesh sieve (Table 1). The same was used to fill the pots. Biochar was mixed with the soil at the rate of 1% on w/w basis, where it was intended as a treatment, before filling the pot. Each pot received 6 kg of soil mixed with N, P, and K at the rates of 120, 60, and 60 kg ha−1, respectively, as recommended for wheat. Surface sterilization of wheat seeds (6 seeds per pot) was performed using a 2% sodium hypochlorite solution for 6 min, followed by rinsing with deionized water. These sterilized seeds were then subjected to a 3-hour treatment with a bacterial culture with optical density of one measured at 600 nm on a spectrophotometer (Shimadzu) with CFU 109 composed of the selected strains. Both the inoculated and non-inoculated seeds were sown equidistantly within each pot. Two levels of soil moisture were maintained throughout the plant growth and development using gravimetric measurements. Water holding capacity (WHC) of the soil was determined without or with 1% biochar mixed in it. For optimum moisture level, soil moisture equivalent to ∼75% of WHC was maintained, whereas soil moisture equivalent to 25% of WHC was maintained to simulate drought. Pots were weighed after every two to five days to replenish the moisture lost to evaporation to maintain these moisture levels. A random arrangement of each treatment, with three replications, was ensured under suitable light and temperature conditions in the glass house. Periodic rotation of the pots in their arrangement was implemented.

Table 1 Physicochemical characteristics of soil and biochar.

Name of variables	Soil	Biochar	
Textural class	Sandy loam	ND	
Sand (%)	46 + 5.7	ND	
Silt (%)	33.6 + 6.0	ND	
Clay (%)	20.3 + .4	ND	
EC (dSm-1)	1.360	2.330	
pH	7.9	7.5	
Available K (mg kg-1 d.m.)	108.45	ND	
Available P (mg kg-1 d.m.)	11.40	491	
Total Na+ (mg kg-1 d.m.)	ND	6.40	
Total K+ (mg kg-1 d.m.)	ND	279	
Total Ca (mg kg-1 d.m.)	ND	401	
Total P (mg kg-1 d.m.)	ND	630	
Total N (mg kg-1 d.m.)	ND	1914.7	

Plant analysis

Plants were harvested at base at maturity and various growth and yield parameters were recorded. Plant height was measured from the ground level to the apex of the topmost leaf while the plants were still alive, and number of spikes and tillers were tallied. After uprooting and gently rinsing, the roots were spread on table and measured from the base to the longest end. For grain yield, grains were individually separated from straw and weighed. The above ground plant weight minus the grains was recorded as straw yield.

Nitrogen content in plant parts was determined by using Kjeldahl method. To determine the nitrogen content, 1 gram of the plant biomass was digested with 10 milliliters of concentrated sulfuric acid (H2SO4) and 5 g of a catalyst mixture in a digestion tube. After cooling, the mixture was processed for distillation. The distillate was collected and titrated against an H2SO4 blank. The total nitrogen content was then calculated from the blank and sample titration readings.

To estimate the phosphorus content, plant P was extracted using 0.5 N sodium bicarbonate (NaHCO3) at pH 8.5, and then treated with ascorbic acid in an acidic medium. The intensity of the blue color produced was measured, and the amount of P was then calculated using spectrophotometer, a standard calibration curve for P by Watanabe & Olsen (1965). To estimate the potassium content, 25 mL of ammonium acetate solution was added to 5 g of the biomass sample. The mixture was shaken for 5 min and then filtered. The amount of K was then measured in the filtrate. To estimate the sodium content, 1 gram of the plant extract was mixed with 80 mL of 0.5 N hydrochloric acid (HCl) for 5 min at 25 °C. The concentrations of these elements were then measured using a flame photometer in the resulting filtrate.

Soil physicochemical analysis

After uprooting the plants from pots at harvest, the remaining soil was thoroughly mixed to make a composite sample. A composite sample weighing a few hundred grams was then stored in refrigerator (4 Co) or a freezer (−20 Co) for subsequent analyses.

Soil pH was measured using a pH meter (Model 671P, Jenway, Sheun Wan, Hong Kong) in a 1:1 soil-to-water suspension. For measuring electrical conductivity (EC), a 1:5 (w/v) soil-water slurry was prepared after shaking it end over end for one hour before measuring the EC using an EC meter (Hannah Instruments, Smithfield, RI, USA).

Microbial biomass carbon (MBC) was determined using fumigation extraction method (Vance, Brookes & Jenkinson, 1987). A 10 g moist soil sample was placed in an open top crucible and fumigated with 30 mL of alcohol-free chloroform (CHCl3) in a desiccator. The fumigated as well as non-fumigated soil samples were then mixed with 50 mL of 0.5 M potassium sulfate (K2SO4) solution and shaken on a horizontal shaker for 30 min at a speed of 200 rpm. The samples were then filtered and the filtrates were digested with potassium dichromate and remaining digester was titrated against acidified ferrous ammonium sulfate following a modified Walkley-Black method (Walkley & Black, 1934; Kaneez-e Batool et al., 2016).

An air dried 2.5 g soil sample was extracted with a 25 mL of a 1 M ammonium acetate (CH3COONH4) solution. The Na+ and K+ concentration in the diluted filtrate was determined by Jenway PFP-7 flame photometer (Method 11a, US salinity Lab. Staff, 1954). Plant available phosphorus was determined using spectrophotometer (Milton Roy, Ivyland, PA, USA) at 880 nm wavelength, following the standard curve method by Watanabe & Olsen (1965). For this purpose, a 2.5 g of the air-dried soil sample was mixed with 25 mL of a Bray-1 extracting solution, which consists of 0.03 M ammonium fluoride (NH4F) and 0.025 M hydrochloric acid (HCl). After 5 mins of shaking, the mixture was filtered through a Whatman No. 42 filter paper to obtain the filtrate containing plant available P. An aliquot of 2 mL of filtrate and 8 mL of molybdate-ascorbic acid reagent were mixed and kept on shelf for 20 min to develop a blue color complex.

To extract the nitrate, 50 mL of a 2 M calcium chloride (CaCl2) solution was added to the 10 g soil sample followed by shaking for 30 min. The suspension was then filtered through a Whatman No. 42 filter paper. The nitrate concentration was then measured by the method of salicylic acid (Cataldo et al., 1975). Ammonium content was measured by using the Indophenol blue method (Kandeler & Gerber, 1988).

Activities of β-glucosidase, chitinase, acid phosphatase and leucine-aminopeptidase were assayed using fluorogenically labeled substrates MUF-ß-D-glucopyranoside (EC 3.2.1.21), MUF-N-acetyl-ß-D-glucosaminide dehydrate (EC 3.2.1.52), MUF-phosphate monoester (EC 3.1.3.2), and L-Leucine-7-amino-4-methylcoumarin (EC 3.4.11.X) respectively (Pritsch et al., 2004; Sanaullah et al., 2011). Briefly, 0.5 g of fresh soil sample was shaken for 30 mins in a 50 mL autoclaved water to prepare a suspension. An aliquot of 50 µL from these suspensions were added to 96-well microplate, where 50 µL of a buffer solution (MES or Trizma) were added to maintain a pH of 6.0 to 6.5. Then, 100 µL of the 200 µM of the specific substrate was added to each well making a total volume of 200 µL. The fluorescence of these suspensions was measured after 2 h at an excitation wavelength of 360 nm and an emission wavelength of 460 nm on a microplate reader (SYNERGY-HTX, BioTek, CA, USA). The enzyme activities were described as nano mol of MUF or AMC released per g of dry soil per hour (nmol MUF/AMC g−1 hr−1).

Statistical analysis

The effect of moisture level, biochar and PGPR on plant growth and yield parameters and soil quality variables were assessed by applying a three-way analysis of variance (ANOVA) (Steel & Torrie, 1980) followed by least significance difference (LSD) at 95% confidence interval to differentiate the means. These analyses were carried out using Statistix 8.1 (Analytical Software, McKinney, TX, USA).

Results

Selection of strains based on PGPR traits and their molecular characterization

A total of 66 strains of rhizobacteria were isolated. Among these, we identified seven isolates, namely BJK1, C4, C5, C14, C20, C25, and C28, that were capable of substantial phosphate solubilization under severe drought conditions (3% PEG 6000). From this group, three strains - BJK1, C4, and GT2 -were chosen for subsequent experiments due to their highest phosphate solubilization index and their potential for producing IAA (Table 2).

Table 2 Biochemical and molecular analysis of rhizobacterial isolates.

Strains	Biochemical analysis	Molecular analysis	Accession number	
	Phosphate solubilization index (PSI)	IAA activity (mg L−1)			
BJK1	6	10.26 ± 0.8	Bacillus thuringiensis	MT292104	
C4	5.7	2.01 ± 0.31	Bacillus tropicus	OM049396.1	
GT2	8.6	6.24 ± 0.25	Bacillus thuringiensis	MN044865.1	

Construction of phylogenetic trees from the genetic sequences of the 16S rRNA indicated that the chosen isolates belonged to the Bacillus genus (Fig. 1). The genetic makeup of BJK1, C4, and GT2 isolates displayed a substantial 100 and 99% similarity to Bacillus subtilis, and these genetic profiles were subsequently recorded in GenBank (Table 2). In the case of isolate BJK1, it exhibited a 99% genetic resemblance to Bacillus thuringiensis (B. thuringiensis) and was thus included in GenBank with accession number MT292104. Isolate C4 was confirmed as Bacillus tropicus (B. tropicus) and was allocated GenBank accession number OM049396.1. Similarly, isolate GT2 was identified as Bacillus paramycoides (B. paramycoides) and was attributed a GenBank accession number MN044865.1.

Wheat growth and yield

Drought stress significantly decreased plant height in uninoculated as well as all the inoculated plants (Fig. 2A). Biochar addition in uninoculated water-stressed pots could not improve it. However, addition of biochar at 75% WHC significantly increased plant height by 8.21% than 75% WHC without biochar (Fig. 2A). Sole applications of B. thuringiensis, B. tropicus and B. paramycoides significantly increased plant height by 10, 11 and 13% respectively under drought stress as compared to the control without biochar at 25% WHC. Co-application of any of these PGPRs with biochar didn’t change plant height compared to sole application of the PGPRs, though the highest plant height was found where biochar was co-applied with B. thuringiensis (i.e., 67 ± 0.66 cm).

Figure 1 Phylogenetic tree of B. thuringiensis, B. tropicus and B. paramycoides.

Figure 2 Response of plant growth and yield variables to biochar and PGPR.

Effect of biochar and three PGPR on plant height (A), root length (B), number of tillers (C), number of spikes (D), straw yield (E), 100-grain weight (F), and grain yield (G) of wheat in optimum and limited moisture conditions. Error bars represent standard errors of means (n =3). Letters on top of bars represent significant difference, at 95% confidence interval, among means based on three-way analysis of variance.

Like plant height, drought significantly decreased root length in uninoculated as well as inoculated plants compared to respective 75% WHC plants (Fig. 2B). Biochar addition at 75% WHC significantly increased root length by 8%, whereas no improvement was observed at 25% WHC. Inoculation with B. thuringiensis, B. tropicus, and B. paramycoides increased root length by 6, 11, and 9% at 75% WHC and by 13, 14 and 15% respectively at 25% WHC. Combining biochar with B. thuringiensis and B. paramycoides further increased root length by 23% and 25% at 75% WHC. However, no significant increase was found when B. tropics was co-applied with biochar. Addition of biochar at 25% WHC with B. thuringiensis, B. tropicus, and B. paramycoides increased root length by 29, 11 and 28% compared to the control at 25% WHC indicating an interaction between biochar and PGPR (Table 3).

Table 3 P values at ≥ 95% confidence interval of two-way analysis of variance with biochar, PGPR, biochar × PGPR and Biochar*PGPR* WHC as factors.

The bold numbers under a factor represent significant effect of that factor on the relevant dependent variable.

Dependent variable	Biochar	PGPR	Biochar*PGPR	Biochar*PGPR* WHC		
Plant height	0.001	0.000	0.000	0.050		
Root length	0.001	0.000	0.000	0.175		
No. of tillers/pot	0.00	0.000	0.004	0.561		
No. of spikes/pot	0.411	0.833	0.000	0.035		
Straw yield/pot	0.004	0.000	0.000	0.515		
100-grain weight	0.044	0.000	0.005	0.233		
Grain yield/ha	0.001	0.824	0.000	0.021		
Grain total P	0.000	0.000	0.768	0.000		
Straw total P	0.001	0.000	0.334	0.169		
Roots total P	0.000	0.000	0.000	0.000		
Grain total N	0.000	0.559	0.000	0.885		
Straw total N	0.000	0.756	0.000	0.745		
Roots total N	0.000	0.665	0.001	0.803		
Grains K+/Na+	0.000	0.000	0.000	0.916		
Straw K+/Na+	0.000	0.000	0.000	0.083		
Roots K+/Na+	0.000	0.001	0.000	0.549		
Soil pH	0.114	0.473	0.867	0.952		
Soil EC	0.002	0.000	0.000	0.568		
Soil MBC	0.000	0.000	0.000	0.213		
Soil K+/Na+	0.000	0.060	0.000	0.293		
Plant available P	0.000	0.000	0.220	0.745		
Soil ammonium nitrogen	0.003	0.982	0.000	0.996		
Soil nitrate nitrogen	0.000	0.448	0.094	0.094		
Chitinase activity	0.691	0.000	0.000	0.370		
Acid phosphatase activity	0.560	0.001	0.000	0.227		
Beta-glucosidase activity	0.022	0.007	0.000	0.250		
Leucine aminopeptidase (LAP) activity	0.294	0.591	0.782	0.731		

The number of tillers significantly increased from 11.33 ± 0.66 (the control without biochar) to 13.33 ± 0.57 in B. thuringiensis, 12 ± 0.57 in B. tropicus, and 12.66 ± 0.33 in B. paramycoides inoculated treatments at 75% WHC. Co-application of biochar and B. thuringiensis further boosted tiller count to 15 ± 0.33 (Fig. 2C). However, biochar addition with B. tropicus and B. paramycoides did not induce any significant change in number of tillers as compared to the control with biochar at 75% WHC, nor there was any interaction between PGPR and biochar. At 25% WHC, the number of tillers was significantly reduced by 35% but biochar addition enhanced the number of tillers to 10 ± 0.57 when compared to control without biochar (7.33 ± 0.88). Furthermore, application of B. thuringiensis, and B. paramycoides with biochar at 25% WHC significantly increased number of tillers to 11.33 ± 0.33 and 13.33 ± 0.66 respectively, compared to the control without biochar.

The number of spikes significantly decreased to 5.33 ± 0.33 from 7 ± 0.57 due to drought (Fig. 2D). However, co-application of biochar and B. thuringiensis increased the number of spikes to 6.33 ± 0.33. No significant results were found by sole application of B. tropicus, and B. paramycoides. However, co-application of biochar with these PGPRs increased the number of spikes to 6 ± 0.88 and 7.33 ± 0.33 respectively. Biochar addition at 75% WHC increased number of spikes to 8.66 ± 0.33 from 7 ± 0.57 (Fig. 2D). When applied alone at 75% WHC, B. tropicus exhibited a significantly higher spike count of 11.33 ± 0.33 compared to B. thuringiensis (7.33 ± 0.88) and B. paramycoides (7.66 ± 0.66). Biochar with B. tropicus had no additional effect, while with B. thuringiensis and B. paramycoides, interactive significant results were observed with 9.66 ± 0.33 and 10.33 ± 0.57 spikes, respectively.

Straw yield significantly decreased by 31% under water deficit conditions (Fig. 2E). Sole applications of biochar, B. thuringiensis, B. tropicus and B. paramycoides increased straw yield by 25, 12, 32 and 25% respectively when compared to the control at 25% WHC. Here, combined application of biochar with B. thuringiensis, B. tropicus and B. paramycoides further enhanced straw yield by 37, 26, and 41% compared to the control at 25% WHC without biochar. The inoculation or biochar addition also improved straw yield at 75% WHC. For instance, inoculation with B. thuringiensis, B. tropicus, and B. paramycoides significantly increased straw yield by 30, 20, and 21%, respectively, compared to the control without biochar 75% WHC. Biochar addition at 75% WHC increased straw yield by 20% than the control without biochar. At 75% WHC, combined application of biochar with B. thuringiensis and B. paramycoides further enhanced straw yield by 34 and 25% respectively, but with B. tropicus it decreased straw yield to 7% compared to the control without biochar.

Drought significantly decreased 100-grain weight by 20% (Fig. 2F). However, biochar, B. thuringiensis, B. tropicus and B. paramycoides increased 100-grain weight by 15, 23, 20 and 20% respectively compared to the control at 25% WHC without biochar. Moreover, combined application of biochar with B. thuringiensis, B. tropicus and B. paramycoides further enhanced 100-grain weight by 30, 23, and 36% respectively compared to the control at 25% WHC without biochar. On the other hand, these applications also improved 100-grain weight under 75% WHC moisture though those improvements were not as large. For instance, biochar and B. thuringiensis increased 100-grain weight by 9% each, however inoculation of B. tropicus and B. paramycoides induced more increase in 100-grain weight by 20 and 17% than the control without biochar at 75% WHC (Fig. 2F). Co-application of biochar with B. thuringiensis and B. paramycoides further increased 100-grain weight by 20 and 31% respectively compared to the control without biochar at 75% WHC. However, B. tropicus decreased it to 10% compared to control without biochar at 75% WHC.

Drought significantly decreased grain yield by 39% (Fig. 2G). Biochar addition alone or in combination with PGPRs improved grain yield under drought stress, though only coapplied biochar and B. tropicus increased it significantly by 34%. Sole application of biochar or any of the PGPRs at 75% WHC didn’t improve grain yield. However, B. tropicus and B. paramycoides when coapplied with biochar at 75% WHC significantly increased it by 26 and 38% respectively, compared to control plants at 75% WHC.

Total phosphorus and nitrogen content in wheat plant

Drought decreased total phosphorus (P) content in grain by about 30%, though this decrease was statistically insignificant (Fig. 3A). Addition of biochar and inoculation with B. paramycoides and B. thuringiensis mitigated drought’s negative effect and increased grain P content by 19, 15, and 57%, respectively. However, B. tropicus exhibited a negative impact on total P contents in grains. Remarkably, the combined application of biochar with B. thuringiensis, B. tropicus, and B. paramycoides showed significant interactive effects, leading to a substantial increase in total P levels by 80, 53, and 82%, respectively, compared to the control without biochar under 25% WHC. On the other hand, in 75% WHC moisture conditions, these applications exerted a more positive effect on grain P. For instance, inoculation with B. thuringiensis and B. paramycoides increased grain P by 59 and 35% respectively, whereas inoculation with B. tropicus significantly decreased it (Fig. 3A). However, adding biochar at 75% WHC elevated the grain P level by 30.39% compared to the control at 75% WHC. Notably, the combined application of biochar with B. thuringiensis, B. tropicus, and B. paramycoides at 75% WHC led to further increase in total P levels by 69, 14, and 47%, respectively, compared to the control without biochar under 75% WHC, indicating an interactive effect between PGPR and biochar.

Figure 3 Effect of co-application of biochar and three PGPR on P content in plant parts.

Effect of biochar and three PGPR on P and N content in grain (A, B), straw (C, D), and roots (E, F) of wheat in optimum and limited moisture conditions. Error bars represent standard errors of means (n =3). Letters on top of bars represent significant difference, at 95% confidence interval, among means based on a three-way analysis of variance.

Drought induced a significant drop by 33% in straw P (Fig. 3C). However, biochar, B. thuringiensis, B. tropicus, and B. paramycoides, significantly increased straw P by 47, 61, 2.68, and 84%, respectively. Moreover, when biochar was co-applied with B. thuringiensis, B. tropicus, and B. paramycoides, straw P increased by 89, 74, and 90%, respectively, compared to the control without biochar at 25% WHC. At 75% WHC sole application of B. thuringiensis, B. tropicus, and B. paramycoides increased straw P by 72, 29, and 59%, respectively, whereas sole application of biochar at 75% WHC increased straw P by 16%. Furthermore, co-application of biochar with B. thuringiensis, B. tropicus, and B. paramycoides at 75% WHC increased straw P by 85, 50, and 70%, respectively indicating a synergistic interaction between PGPR and biochar.

Similar to straw and grain P, drought also significantly decreased phosphorus content in wheat roots (root P) (Fig. 3E). However, biochar, B. thuringiensis, B. tropicus, and B. paramycoides increased root P by 49, 65, 51, and 87%, respectively, whereas co-application of biochar with B. thuringiensis, B. tropicus, and B. paramycoides raised it by 74, 70, and 87%, respectively. This interaction highlighted the synergistic impact between PGPR and biochar on total P levels in roots (Table 3). In 75% WHC moisture conditions too, inoculation by PGPRs as well as biochar application significantly increased root P. Briefly, root P increased by 66, 81, 56, and 73%, respectively after addition of biochar, B. thuringiensis, B. tropicus, and B. paramycoides respectively. However, only co-application of biochar with B. thuringiensis significantly increased it further, by 82%, while co-application with B. tropicus, and B. paramycoides decreased compared to sole application of these inoculants.

In corollary to reduced plant growth under drought, N content in grain, straw, and roots also significantly dropped due to drought, by 34, 43 and 6% respectively (Figs. 3D, 3E, & 3F). However, co-application of biochar with B. thuringiensis, B. tropicus and B. paramycoides under drought conditions increased N content by 9, 22 and 17% in grains, by 17, 28 and 14% in straw, and by 3, 4 and 5% in roots respectively. On the other hand, biochar addition increased N level by 15, 23 and 5% in grain, straw, and roots respectively at 75% WHC. However, inoculation with any of the three PGPRs did not induce any change in plant N content at any moisture level. Moreover, co-application of biochar with any of the three PGPRs at 75% WHC did not induce any further increase in plant N, rather it decreased it in certain cases.

K+/Na+ ratio in wheat grain, straw, and roots

Drought did not change K+/Na+ ratio in grain (Fig. 4A). However, biochar, B. thuringiensis, B. tropicus and B. paramycoides enhanced grain K+/Na+ ratio by 21, 22, 7 and 11% respectively, whereas co-application of biochar with B. thuringiensis, B. tropicus and B. paramycoides increased grain K+/Na+ ratio by 39, 38 and 34% respectively. Similar to drought conditions, biochar addition at optimum moisture also increased grain K+/Na+ significantly, by 29%, whereas inoculation with B. thuringiensis, and B. paramycoides increased grain K+/Na+ ratio by 38 and 5% respectively, whereas B. tropicus did not induce any change. However, co-application of biochar with B. thuringiensis, B. tropicus and B. paramycoides at 75% WHC further increased K+/Na+ ratio by 52, 37 and 42% respectively.

Figure 4 K+/Na+ ratio in plant parts.

Effect of biochar and three PGPR on K+/Na+ ratio in (A) grain, (B) straw, and (C) roots of wheat in optimum and limited moisture conditions. Error bars represent standard errors of means (n =3). Letters on top of bars represent significant difference, at 95% confidence interval, among means based on a three-way analysis of variance.

Unlike grain, drought significantly decreased K+/Na+ ratio in straw, by 19.7% (Fig. 4B). Biochar addition boosted it by 24% under drought conditions. Moreover, co-application of biochar with B. thuringiensis, and B. tropicus increased it by 27, and 29% respectively, whereas co-application with B. paramycoides decreased it compared to control at 25% WHC without biochar. On the other hand, inoculation with PGPRs at 75% WHC did not induce any improvement in straw K+/Na+ ratio. However, co-application of biochar with only B. tropicus at 75% WHC increased K+/Na+ ratio by in 17%, compared to the control without biochar at 75% WHC.

Drought significantly decreased root K+/Na+ ratio by 17.75% (Fig. 4C). There were no significant results obtained by inoculation with PGPRs B. thuringiensis and B. tropicus, however B. paramycoides increased it by 13% at 25% WHC compared to the control with 25% WHC. Moreover, co-application of biochar with B. thuringiensis, B. tropicus and B. paramycoides increased root K+/Na+ by 25, 28 and 36% respectively. At 75% WHC moisture conditions, B. thuringiensis, and B. tropicus inoculation significantly increased root K+/Na+ ratio by 9 and 7% respectively, whereas inoculation with B. paramycoides did not show any significant change. Moreover, co-application of biochar with B. thuringiensis, B. tropicus and B. paramycoides at 75% WHC increased root K+/Na+ ratio by 16, 18 and 17% respectively compared to the control without biochar at 75% WHC.

Soil EC, pH, microbial biomass carbon and K+/Na+ ratio

Neither the inoculation nor the biochar addition induced any change in soil pH at any moisture level (Fig. 5A).

Figure 5 Effect of application of biochar and PGPR on soil pH, EC, MBC, and K+/Na+ ratio in soil.

Effect of biochar and three PGPRs on (A) soil pH, (B) EC, (C) soil microbial biomass carbon, and (D) K+/Na+ ratio in soil in optimum and limited moisture conditions. Error bars represent standard errors of means (n =3). Letters on top of bars represent significant difference, at 95% confidence interval, among means based on a three-way analysis of variance.

Biochar addition significantly reduced soil EC at both moisture levels (Fig. 5B). Biochar addition in combination with B. thuringiensis, and B. paramycoides further decreased the soil EC to 1.19 dS m−1, and 1.18 dS m−1 respectively at 75% WHC.

Drought significantly decreased microbial biomass carbon by almost 33% irrespective of biochar addition (Fig. 5C). MBC significantly increased in response to inoculation of all the three PGPRs i.e B. thuringiensis, B. tropicus and B. paramycoides as well as the addition of biochar (Fig. 5C). In response to B. thuringiensis, MBC increased by 33%, whereas its co-application with biochar further increased MBC by 45% as compared to the control without biochar at 75% WHC. Inoculation with B. tropicus and B. paramycoides increased MBC by 46 and 22% respectively, whereas co-application with biochar increased it by 51 and 43%. At 25% WHC moisture level, B. thuringiensis inoculation increased MBC by 22%, whereas its co-application with biochar further increased MBC by 39.9% in total as compared to the control without biochar at 25% WHC. Inoculation with B. tropicus and B. paramycoides increased MBC by 26 and 30% respectively, whereas co-application with biochar increased it by 33 and 45%.

At 75% WHC moisture level, none of the PGPRs changed K+/Na+ ratio, though biochar addition increased it by 46% (Fig. 5D). Moreover, biochar co-inoculation with B. thuringiensis, B. tropicus and B. paramycoides significantly increased K+/Na+ ratio in soil by 43, 48 and 44% respectively than the control at 75% WHC. The soil K+/Na+ ratio was significantly decreased by 32% under drought conditions. Moreover, biochar or inoculation with any of the PGPRs did not improve it. However, co-inoculation of biochar with B. thuringiensis, B. tropicus and B. paramycoides significantly increased K+/Na+ ratio by 44, 41% and 42% respectively than the control at 25% WHC.

Soil nitrogen and phosphorus content

Drought or addition of biochar did not influence plant available P at any moisture level (Fig. 6A). At 75% WHC moisture, B. thuringiensis and B. paramycoides inoculation increased P by 19.52 and 17.55% respectively. Co-application of all three PGPRs B. thuringiensis, B. tropicus, and B. paramycoides with biochar significantly increased available P by 28, 14 and 37% compared to the control without biochar at 75% WHC. Similarly, co-application of biochar with B. thuringiensis and B. paramycoides significantly increased plant available P by 21 and 31% at 25% WHC, whereas co-application with B. tropicus did not induce any change.

Figure 6 Effect of biochar and PGPR application on N and P availability in soil.

Effect of biochar and three PGPRs on (A) plant available phosphorus, (B) soil ammonium, and (C) soil nitrate in optimum and limited moisture conditions. Error bars represent standard errors of means (n =3). Letters on top of bars represent significant difference, at 95% confidence interval, among means based on a three-way analysis of variance.

Sole application of biochar did not induce any change in ammonium content at any moisture level (Fig. 6B). Similarly, ammonium in soil remained unchanged in response to inoculation by any of the three PGPRs at both moisture levels.

At 75% WHC, addition of biochar significantly increased the nitrate content by 6% compared to the control without biochar at 75% WHC (Fig. 6C). Nitrate in soil remained unchanged in response to B. thuringiensis, B. tropicus and B. paramycoides at both 75% WHC and 25% WHC.

Extracellular enzymatic activity in soil

Drought significantly decreased chitinase activity by ∼21% irrespective of biochar addition, while biochar addition at both moisture levels slightly increased it (Fig. 7A). Moreover, chitinase activity slightly increased in response to inoculation with B. thuringiensis and B. tropicus while B. paramycoides decreased it by 26% at 75% WHC. Moreover, co-application of biochar with B. tropicus and B. paramycoides decreased chitinase activity by 12 and 28% respectively. At 25% WHC moisture, B. thuringiensis and B. paramycoides decreased chitinase activity by about 18%, whereas B. tropicus did not influence it. Co-application of biochar with the PGPRs did not further change the chitinase activity.

Figure 7 Effect of B. thuringiensis, B. tropicus and B. paramycoides with and without biochar on extracellular enzyme activity in soil.

Effect of B. thuringiensis, B. tropicus and B. paramycoides with and without biochar on (A) chitinase, (B) acid phosphatase, (C) β-glucosidase, and (D) leucine aminopeptidase. Standard errors of means (n =3) are shown as error bars. Different letters on top of bars show a significant difference among means based on a three-way ANOVA followed by an LSD at 95% confidence interval.

Drought substantially decreased acid phosphatase activity, while biochar addition massively recovered it (Fig. 7B). At 75% WHC moisture level, acid phosphatase activity did not undergo a substantial change in response to inoculation with any of the three PGPRs. However, co-application of biochar with B. thuringiensis, B. tropicus and B. paramycoides decreased acid phosphatase activity by 21, 8 and 24% respectively at 75% WHC. Under drought conditions, co-application of biochar and B. tropicus increased phosphatase activity by 30%, whereas the co-application with the other two PGPRs reduced it significantly.

β-glucosidase activity was also substantially decreased by drought irrespective of biochar addition (Fig. 7C). At 75% WHC moisture, it decreased in response to addition of biochar by 19%, and in response to B. tropicus and B. paramycoides by 20% and 58% respectively. Co-application of biochar with all three PGPRs decreased β-glucosidase activity at 75% WHC. At 25% WHC, B. tropicus and B. paramycoides increased soil β-glucosidase activity by 6.7% and 25% respectively, however, B. thuringiensis further decreased it. Biochar with B. tropicus showed interactive positive effect by increasing soil β-glucosidase activity by 32% at 25% WHC than the control without biochar at 25% WHC.

Drought significantly decreased leucine aminopeptidase activity whereas biochar addition improved it at both moisture levels (Fig. 7D). However, all of three PGPRs showed negative effect on leucine aminopeptidase activity at 75% WHC, whereas co-application with biochar did not change it any further. On the other hand, under drought conditions, B. thuringiensis and B. paramycoides increased it by 15%, whereas no effect was induced by B. tropicus. Moreover, co-application of biochar with any of the PGPRs did not change leucine aminopeptidase activity at 25% WHC.

Discussion

Biochar and drought-tolerant PGPRs have been shown to alleviate drought stress for plants and improve their productivity (Lalay, Ullah & Ahmed, 2022; Zulfiqar et al., 2022). We hypothesized that their co-application would have a more positive effect, if not synergistic, on stress alleviation and productivity.

Effect of PGPRs on wheat productivity and soil health

The PGPRs that were finalized for pot experiment based on their potential to produce indole acetic acid, solubilize rock phosphate and tolerate drought coincidentally belonged to Bacillus genus (Table 2). This group of PGPRs have been well documented to help alleviate drought stress for wheat plants in addition to directly promoting the latter’s growth (Lastochkina, 2021). True to their reported potential, inoculation of wheat with the three PGPRs i.e., B. thuringiensis, B. tropicus, and B. paramycoides alleviated drought stress and enhanced wheat growth and productivity. For instance, they increased plant height by 10%, 11%, and 13%, respectively and root length by 13%, 14% and 15% compared to uninoculated pots under drought conditions (Fig. 2A). The ability of Bacillus sp. to enhance plant height and root length under water-limited conditions can be attributed to their diverse plant growth-promoting mechanisms. One of the primary mechanisms by which Bacillus species can improve them is through the production of phytohormones, such as IAA and cytokinin (Vurukonda et al., 2016). These plant hormones play a crucial role in stimulating cell division and elongation, which can directly contribute to increased plant length (Sosnowski, Truba & Vasileva, 2023). Additionally, Bacillus species have been reported to enhance root growth and development, which can improve the plant’s ability to acquire water and nutrients from the soil, even under drought stress conditions (Marasco et al., 2012; Sandhya et al., 2010). Improved root systems can lead to better anchorage, increased water and nutrient uptake, and ultimately, taller plant growth. Inoculation with all the three PGPRs induced a significant increase in the number of tillers. However, this increase in tillers didn’t translate into a significant higher number of spikes in B. tropicus and B. paramycoides inoculated pots. PGPRs induced stimulation in various growth-related wheat parameters significantly increased straw yield by 12–32%. B. thuringiensis and B. paramycoides inoculation increased the grain yield, though this increase was not significant. It may happen that the improvement in biological traits may not translate into a significant improvement in grain/seed yield particularly under abiotic stress conditions (Kaushal & Wani, 2016; De Andrade et al., 2023).

All the three PGPRs used in this study were strong solubilizers of phosphate (Table 1). This is also reflected in significantly higher P in almost all parts of wheat plants even under drought. Moreover, a 19.52 and 17.55% higher plant available P in B. thuringiensis and B. paramycoides inoculated pots respectively, at 75% WHC moisture indicates the strength of these PGPRs that they were solubilizing more than the plants required (Fig. 6). However, B. tropicus at both moisture levels while the other two PGPRs at 25% WHC, despite improving the P uptake by plants, did not solubilize so abundantly that the available P in soil could be higher than the relevant control. Under circumstances where a mineral nutrient is sufficiently available, soil microorganisms tend to disinvest in the secretion of that nutrient-specific enzyme in order to allocate the precious resources elsewhere (Sinsabaugh et al., 2008). Consequently, the treatments where available P was higher or sufficient for plant (and microbial) uptake, acid phosphatase activity was lower or did not change compared to the respective control (Fig. 7B). Although the PGPRs didn’t change availability of mineral nitrogen, ammonium or nitrate in soil, they altered the activities of N cycling enzymes chitinase and leucine aminopeptidase (Figs. 6 & 7). They slightly but significantly decreased chitinase but substantially increased leucine aminopeptidase activity particularly under drought conditions. The latter must be to fulfill the microbial demand for more mineral N given that the P was becoming available in the soils due to phosphate solubilizing activity of PGPRs thereby leading to other soil microbes to invest in N acquiring enzymes. An increase in microbial biomass in PGPRs inoculated pots corroborates this conclusion since the microbes can only grow in mass when they have enough N to meet their stoichiometric need when enough P is available (Kirkby et al., 2011; Shahzad et al., 2019).

Effect of biochar on wheat productivity and soil health

Biochar improves water use efficiency in coarse textured soils thereby improving the plant productivity, whereas the same is not effective in improving hydraulic properties of fine textured soils (Atkinson, 2018; Edeh, Mašek & Buss, 2020). The soil used in this study was relatively fine textured soil. Therefore, we did not determine the effect of biochar on soil hydraulic properties and assessed biochar’s effect on plant productivity to indirectly determine its drought-stress alleviating potential. Biochar is a nutrient rich substrate that is expected to raise mineral nutrients content in soil and promote plant uptake of them. Hence, biochar addition significantly increased the nitrate content at optimum as well as low moisture conditions (Fig. 6). Biochar becomes concentrated with nutrients during pyrolysis thereby becoming richer in nutrients compared to its feedstock (El-Naggar et al., 2019; Knoblauch et al., 2021). On the other hand, biochar did not improve available soil P content at any moisture level (Fig. 6). However, the acid phosphatase activity was significantly higher after biochar addition under drought conditions (Fig. 7) indicating a quest of microorganisms for acquiring P (Stock et al., 2019). Although plant available P wasn’t significantly higher after biochar addition, the plant P uptake was indeed significantly higher at both levels as was the N uptake. The improvement in plant nutrition resulted in significant increase in plant tillers, grain weight and straw yield thereby alleviating the drought stress (Fig. 2). Although biochar did not increase grain yield significantly, there was a noticeable increase in it. These results prove our first hypothesis, which stated that the biochar alone is capable of alleviating drought stress on plants by improving the soil health indicators. Indeed, biochar is known for improving plants’ ability to counter the deleterious effects of drought by increasing their nutrition (Kammann et al., 2011; Mulcahy, Mulcahy & Dietz, 2013; Zhou et al., 2015). Moreover, biochar has also been reported to improve soil physical, chemical and biological properties even under moisture deficit conditions, which consequently may improve plant productivity (Edeh, Mašek & Buss, 2020; Zaheer et al., 2021; Malik et al., 2022).

Effect of co-applied biochar and PGPRs on wheat productivity and soil health

The co-application of phosphate solubilizing and IAA producing three PGPRs used in this study alongside biochar improved soil health substantially better than their individual application. Further confirming our second hypothesis, these improvements in soil health resulted into a higher wheat productivity compared to sole applications of biochar and PGPRs. For instance, microbial biomass was significantly higher in all co-applied treatments under drought, whereas sole inoculation with B. tropicus, and B. paramycoides or application of biochar did not improve it at all (Fig. 5). The increased MBC can contribute to improved soil fertility, water-holding capacity, and ecosystem functioning, particularly in drought-prone regions (Bogati & Walczak, 2022). Though, it is intriguing that biochar itself did not increase microbial biomass despite being a rich source of nutrients and capable of providing conducive habitats to microbial growth (Riaz et al., 2017; Hossain et al., 2020). Apparently, biochar in this soil didn’t increase any of the two macronutrients i.e., N & P. Alternatively, plant uptake of N and P was indeed significantly higher upon application of biochar only indicating that biochar at least facilitated uptake or increased the nutrient just enough to increase the plant nutrient content (Fig. 3). However, the combined application of biochar and P-solubilizing PGPRs released P from biochar and increased its content in soil in addition to increasing its uptake by plants (Fig. 6). In addition to improving microbial biomass and P availability, co-application of biochar and PGPRs substantially lowered electrical conductivity and improved potassium to sodium ratio in soil. Both of these variables play a crucial role in determining the plant’s ability to uptake mineral nutrients from soil, particularly mineral nitrogen (Akhtar et al., 2015; Fazal & Bano, 2016). Indeed, even if the co-application did not further increase the mineral nitrogen availability in soil, the higher uptake of the same plants can be explained by this indirect mechanism.

Overall results indicate that the co-applied PGPRs and biochar worked in tandem through various mechanisms to complement and enhance each other’s positive effect on soil health improvement and wheat productivity. Firstly, PGPRs solubilized P from biochar thereby leading to higher uptake by plants, higher growth and yield. Secondly, PGPRs induced growth in plant roots and shoots which effectively increased the plants’ ability to uptake the nutrients and water thereby increasing the use efficiencies of both (Figs. 2 & 3). Though we didn’t measure, we can presume that biochar application must have improved soil’s physical environment thereby facilitating better microbial activity. The improvements in soil health indicators translated into enhanced root and shoot length, number of spikes, tillers, straw yield, grain weight and grain yield.

Conclusion

Biochar and the three PGPRs used in this study i.e., Bacillus thuringiensis, Bacillus tropicus, and Bacillus paramycoides generally improved wheat growth and yield under drought conditions when applied separately. However, the co-application of PGPR and biochar resulted in higher improvement in wheat growth and yield and soil health. These two types of additives worked in tandem via multiple direct and indirect mechanisms. The PGPRs capable of releasing growth stimulating hormones on one hand and solubilizing phosphate on the other helped plants overcome drought and grow better. While the biochar being a reservoir of nutrients on one hand and conditioner of improving soil’s chemical and physical properties on the other provided a suitable environment to microorganisms including the applied PGPRs thereby increasing their activity and efficacy. Our study demonstrated that co-application of biochar and PGPRs capable of releasing growth stimulating hormones and releasing locked nutrients in soil is an excellent integrated approach to alleviate drought stress for plants.

Supplemental Information

Supplemental Information 1 A picture of the pots grown for the experiment

Supplemental Information 2 Raw data of sequences of the three PGPRs used in the study

Supplemental Information 3 Raw data

Additional Information and Declarations

Competing Interests

Author Contributions

DNA Deposition

Data Availability

Tanvir Shahzad and Sabir Hussain are Academic Editors for PeerJ.

Laraib Malik conceived and designed the experiments, performed the experiments, analyzed the data, prepared figures and/or tables, authored or reviewed drafts of the article, and approved the final draft.

Sabir Hussain conceived and designed the experiments, analyzed the data, authored or reviewed drafts of the article, and approved the final draft.

Muhammad Shahid conceived and designed the experiments, analyzed the data, authored or reviewed drafts of the article, and approved the final draft.

Faisal Mahmood performed the experiments, authored or reviewed drafts of the article, and approved the final draft.

Hayssam M. Ali conceived and designed the experiments, analyzed the data, authored or reviewed drafts of the article, and approved the final draft.

Mehreen Malik performed the experiments, prepared figures and/or tables, authored or reviewed drafts of the article, and approved the final draft.

Muhammad Sanaullah performed the experiments, authored or reviewed drafts of the article, and approved the final draft.

Zubda Zahid performed the experiments, analyzed the data, prepared figures and/or tables, authored or reviewed drafts of the article, and approved the final draft.

Tanvir Shahzad conceived and designed the experiments, analyzed the data, authored or reviewed drafts of the article, and approved the final draft.

The following information was supplied regarding the deposition of DNA sequences:

The sequences are available at NCBI: MT292104, OM049396.1, MN044865.1.

The following information was supplied regarding data availability:

The raw values of all the variables as graphs/tables are available in the Supplemental Files.

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
