# Peer review of "Co-applied biochar and drought tolerant PGPRs induced more improvement in soil quality and wheat production than their individual applications under drought conditions"

_PeerJ, doi:10.7717/peerj.18171_

## Round 0.1 · original submission · Major Revisions

Dear Dr Shahzad,
Your work has been evaluated by four independent experts. All of them agreed that this work could be published in PeerJ, but it needs significant improvement beforehand. The reviewers particularly noted that the Materials and Methods section must be supplemented by additional information. On fig. 2 error bars are missing. Please carefully review all reviewers' comments and address them accordingly. We have noticed that reviewers request the inclusion of some publications in the cited literature by you. Of course, you can do so, but if you believe that these publications cannot be cited by you, it will not affect the publication decision in any way.

With best regards,

**Language Note:** The review process has identified that the English language must be improved. PeerJ can provide language editing services - please contact us at [email protected] for pricing (be sure to provide your manuscript number and title). Alternatively, you should make your own arrangements to improve the language quality and provide details in your response letter. – PeerJ Staff

Reviewer 1 ·

Basic reporting

MNS is clear with good english well cited literature and other requiremnent

Experimental design

added in pdf comments

Validity of the findings

valid

Annotated reviews are not available for download in order to protect the identity of reviewers who chose to remain anonymous.

·

Basic reporting

Dear Editor,
I would like to thank you for your confidence in reviewing this manuscript.

Type of manuscript: Article

Title: Co-applied biochar and drought tolerant PGPR induced more improvement in soil quality and wheat production than their individual applications under drought conditions.
Abstract:
Line 34: Replace “(75%WHC) & induced drought (25% WHC)” by “(75% WHC) and induced drought (25% WHC)”. What does mean “WHC)”? Also, change “&” in all the main text with “and”.
Line 37: What does “at the end of the experiment”? Did you want to say at harvest? Please, change or reformulate it for more understanding.
Line 39: Add coma before “especially”.
Line 40: Replace “grian yield by 10% compared to control” by “grain yield by 10% compared to the control”.
Line 42: Which information represents “it”? Please, verify.
Line 43: Did you want to say “plant available phosphorus” or “soil available phosphorus”?
Line 45: Replace “to control” by “to the control”. Same remark in all the main text.
Why authors did not add keywords?

Introduction:
Line 74: Please, check the citation format.
Add These references to support drought and PGPR:
https://doi.org/10.1007/s10343-022-00651-0
https://doi.org/10.3390/su142315984
https://doi.org/10.3389/fpls.2020.516818

M&M:
Line 114: Replace “ml” by “mL”. Same remark in all the main text.
Line 116: Replace “1°C” by “1 °C”. Same remark in all the main text.
Line 118: Please, check the citation format. Same remark in all the main text.
Line 148: Which Korea? South or North?
Line 188: Replace “600nm” by “600 nm”. Same remark in all the main text for this unit and others units except %.
Applied treatments missed in the M&M section, please, provide it.

Results:
All figure and table abbreviations need to be explained.
Could the authors provide some photos of their experience?

References:
Please, check all references citation and list.
Species scientific names in italics.
Some references seem uncompleted.

Experimental design

M&M:
Line 114: Replace “ml” by “mL”. Same remark in all the main text.
Line 116: Replace “1°C” by “1 °C”. Same remark in all the main text.
Line 118: Please, check the citation format. Same remark in all the main text.
Line 148: Which Korea? South or North?
Line 188: Replace “600nm” by “600 nm”. Same remark in all the main text for this unit and others units except %.
Applied treatments missed in the M&M section, please, provide it.

Validity of the findings

Good

Additional comments

References:
Please, check all references citation and list.
Species scientific names in italics.
Some references seem uncompleted.

Reviewer 3 ·

Basic reporting

The manuscript is clear and no major ambiguity found in the text.

Experimental design

The experimental design is clear

Validity of the findings

No comments

Additional comments

"This paper provides a strong foundation for the study, effectively framing the research within the context of global food security, drought stress, and the potential benefits of PGPR and biochar. However there is need to improve some minor changes.
Line no. 30: Remove the word "uncultivated".
Line 32: Replace "plant growth promoting rhizobacteria" with the abbreviation PGPR.
Line 40: Insert a comma after "15%".
Line 60 and 61: Replace "caused" with "causes" and "affects" with "affecting".
Line 108: Add a hyphen between "drought" and "tolerant" to read "drought-tolerant".
Line 137: Remove the redundancy of "a period of".
Line 161: Add "a" before "duration of 2 days".
Line 163: Replace "prior to" with "before".
Line 188: Correct the spelling of "spectophometer" to "spectrophotometer".
Line 230: Correct the agreement mistake by changing "concentration" to "concentrations".
Line 259: Remove the repeated phrase "respectively".
Line 280: Remove the dot from the start.
Line 291: Add a comma after "B. tropicus" and include it throughout the document.
Line 296: Correct the verb "decreases" to "decreased".
Line 325: Correct the spelling of "whileIincorporating" to "while incorporating".
Line 341: Insert a space between "increaseof" and "asynergistic"
Moreover, authors are advised to include this recent literature in introduction and discussion section

(https://doi.org/10.1007/s12517-022-10876-y; https://doi.org/10.3389/fpls.2022.950944; https://doi.org/10.1038/s41598-023-50623-1; https://doi.org/10.1007/s10343-023-00845-0; https://doi.org/10.1016/j.stress.2023.100333; https://doi.org/10.1016/j.stress.2023.100261)
There may be spell and grammatical mistakes creeping within the text, authors must check and correct them.

Reviewer 4 ·

Basic reporting

The manuscript entitled “Co-applied biochar and drought-tolerant PGPR induced more improvement in soil quality and wheat production than their individual applications under drought conditions” investigates the synergistic utilization of biochar and drought-tolerant plant growth-promoting rhizobacteria (PGPR) to mitigate drought stress in wheat cultivation. It underscores that drought stress severely affects plant growth and crop productivity. To address this issue, the application of PGPR or biochar is proposed as a means to enhance plant growth under drought conditions. PGPR strains were isolated from arid uncultivated soils and characterized for their growth-promoting properties. Wheat was subjected to normal and induced drought conditions, with biochar and PGPR applied individually and in combination. Findings reveal that biochar supplementation notably improved wheat growth parameters, particularly under stress conditions. Moreover, the combined application of biochar with specific PGPR strains demonstrated synergistic effects, further enhancing plant growth and soil quality. This collaborative approach was observed to alleviate drought stress by improving nutrient availability and absorption, ultimately fostering wheat growth in challenging environments. The study underscores the potential of co-applying biochar and drought-tolerant PGPR for enhancing soil quality and wheat production under drought conditions, offering a promising avenue for sustainable agricultural practices. However, there are a few points that need clarification or further details. The introduction section overly emphasizes crop production, with minimal background on the published applications of PGPR and Biochar. Why did the author choose Bacillus thuringiensis, Bacillus tropicus, or Bacillus paramycoides? And the full name of PGPR should be written the first time it appears in the introduction.

Experimental design

1. Line 157-163: Suggest adding more details, such as incubation method for the chosen strains and the concentration of cultures used for inoculation of wheat seeds. What’s NA media?

Validity of the findings

1. I suggest attaching the sequence of the three PGPRs obtained from 16s sequencing result, how much accuracy of the identity? How 16S sequencing result is analyzed is not fully described in Methods.
2. It’s better to adjust the aspect ratio and resolution of the photo for phylogenetic tree. Format of strain names is not consistent.
3. Figure 2g: error bar is missing. It’s better to re-align some letters in figure 2 showing significant difference.

Additional comments

1. Line 34: “75%WHC” should be “75% WHC).
2. Line 39-40, 43-44: suggest rephrasing for an improved grammar and clarity.
3. Line 121: which assay is included for “the following criteria”? Please improve clarity.
4. Figure 3: “N content roots of wheat: should be “N content in roots”. “Three PGPR” should be “Three PGPRs”.
5. Line 280 and 339, 374: “. Biochar” and “Fig. Biochar”?
6. Please recheck the format of references, DOI is missing for partial references.
7. Line 420: “all three the PGPRs”?
8. Line 459: please rewrite for improved grammar and clarity.

---

## Round 0.2 · accepted · Accept

Dear Dr Shahzad,

Your work has been re-evaluated by 2 independent experts. Both of them agreed that this work could be published in PeerJ in current version. My congratulations!

With best regards,

Reviewer 1 ·

Basic reporting

Paper can be accepted in present form

Experimental design

Well designed

Validity of the findings

Findings are valid

Additional comments

NA

Reviewer 4 ·

Basic reporting

NA

Experimental design

NA

Validity of the findings

NA

Additional comments

Based on revision of manuscript, the authors successfully incorporated the point to point wise through the manuscript as suggested comments so that I would like to strongly recommend that acceptance of manuscript in your prestigious journal for publication. Thanks!